# Large Scale Flood Hazard Analysis by Including Defence Failures on the Dutch River System

**Alex Curran** [1,2,*] 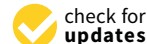**, Karin M. de Bruijn** [2]**, Wouter Jan Klerk** [1,2] **and Matthijs Kok** [1]

[1]  Faculty of Civil Engineering and Geosciences, TU Delft, 2628 CN Delft, The Netherlands
[2]  Department of Flood Risk Management, Deltares, 2600 MH Delft, The Netherlands
*  Correspondence: a.n.curran@tudelft.nl

**Abstract:** To make informed flood risk management (FRM) decisions in large protected river systems, flood risk and hazard analyses should include the potential for dike breaching. 'Load interdependency' analyses attempt to include the system-wide effects of dike breaching while accounting for the uncertainty of both river loads and dike fragility. The intensive stochastic computation required for these analyses often precludes the use of complex hydraulic models, but simpler models may miss spatial inundation interactions such as flows that 'cascade' between compartmentalised regions and overland flows that 'shortcut' between river branches. The potential for these interactions in the Netherlands has previously been identified, and so a schematisation of the Dutch floodplain and protection system is here developed for use in a load interdependency analysis. The approach allows for the spatial distribution of hazard to be quantified under various scenarios and return periods. The results demonstrate the importance of including spatial inundation interactions on hazard estimation at three specific locations, and for the system in general. The modelling approach can be used at a local scale to focus flood-risk analysis and management on the relevant causes of inundation, and at a system-wide scale to estimate the overall impact of large-scale measures.

**Keywords:** flood-risk management; dike breaches; load interdependencies; hazard analysis; hydraulic modelling

## 1. Introduction

### 1.1. Load Interdependencies

In the flood hazard analysis of protected lowland river systems, the spatial and temporal changes in local and system-wide hazard due to dike breaches are often called 'load interdependencies' [1], 'river system behaviour' [2,3] or 'system-risk' [4]. Analysis of this behaviour has become more widespread in recent years, thanks, in part, to a 'systems approach' [5] to Flood Risk Management (FRM) being adopted. Examples of load interdependency analysis in FRM include those by Ciullo et al. [6] and Dupuits et al. [7,8], who demonstrated the importance of this behaviour when developing optimal management strategies.

Load interdependency analyses have been used to estimate flood impacts such as: fatalities [9], economic losses [10], and both economic losses and fatalities [11]. However, much of the research into the field has focused on accurately identifying changes in hazard rather than flood risk (for example, see the papers by the authors of [12,13]). In either case, the occurrence of breaches in space and time usually causes the greatest uncertainty, and therefore a probabilistic Monte Carlo method is almost universally adopted [4,14]. In the Monte Carlo approach, within any given simulation of the system, the occurrence of a breach is deterministic and dependent on hydraulic loads and dike strengths sampled from distributions.

### 1.2. Spatial Aspects of Load Interdependencies

In a load interdependency analysis, the location of a potential breach is usually based on a discretisation of the dike system, to which distributions of dike strength such as fragility curves or functions (for example, see the papers by the authors of [15,16]) are applied. This discretisation can be determined by distance, for example on the Po River (1.2 km sections [17]) and the Elbe River (500 m sections [18]), or on similarities in inundation consequences (e.g., Rhine River [19]). The dependency of failure between neighbouring sections can also be considered, for example by Assteerawatt et al. [10] and in the Dutch Hydra-Ring software [20].

Once a breach occurs, downstream river flow is reduced. This aspect of system behaviour has been termed 'a positive interdependency' [1,7,21] due to the reduced hazard and risk downstream. The breach outflow over time must be accurately calculated in order to quantify both the remaining discharge in the river system and the inundation to the connected floodplain. This outflow will be a function of the river discharge and the floodplain topography but is also heavily influenced by the breach growth in time. Empirical methods to estimate breach growth are available [22], however in many load interdependency analyses simplified growth functions are used.

Various methods exist to model the effect of dike breaches on river flows and floodplain inundation, as discussed by Klerk [1]. Breach simulations often use a fixed grid or mesh where flow is modelled using the two-dimensional (2D) St. Venant equations in the floodplain domain [23]. However, such fully hydrodynamic models are not widely used in load interdependency analyses, primarily due to the computational effort required when running Monte Carlo simulations. Nevertheless, load interdependency analysis using a fixed grid in the 2D domain can be performed for relatively small systems or subsystems of embanked rivers [17,24]. Fast 2D models that solve simplified versions of the St. Venant equations [25,26] have also been used in load interdependency analyses [4,27].

Floodplains can also be schematised in one-dimension (1D) based on expected flow routes (see, for example, the work by the authors of [28]) and can perform as well as 2D simulations for certain topographies [29]. Examples of 'zero-dimensional' (0D) schematisations (which split up the floodplain into different areas with a certain storage capacity) include the RFSM model [26], used in a load interdependency analysis by Gouldby et al. [4]. In software packages where this modelling type is not explicitly implemented, it can be approximated using connected retention nodes. An example of this is Sobek [30], which has been used with estimates of polder storage capacities in various load interdependency analyses [3,12,15]. In these studies, storage nodes were used at each breach location, but the potential for dynamic interactions between these storage volumes (as modelled by Klerk [1]) was not implemented. The use of either 0D or 1D floodplain modelling necessitates calibration against 2D simulations, or where possible, against observed inundation extents and depths.

With a sufficiently accurate schematisation of the floodplain domain in a load interdependency analysis 'negative' interdependencies can be modelled. In compartmentalised floodplain regions these interdependencies have the potential to occur when flood volumes move between the regions (cascading) or return to the river system through the compartment (shortcutting). In either case, the effect of the breach to hazard in downstream areas has increased, thus causing the 'negative' interdependency between components. While this effect has the potential to occur in any protected river system, the low-lying, branched and compartmentalised river system of the Netherlands is particularly vulnerable, as discussed in Section 2. Therefore, in the present study, a load interdependency analysis for the Netherlands is combined with 'fast' quasi-2D modelling methods to produce hazard estimates under various defence failure scenarios relating to breaches caused by floodplain inundation. The introduction of the fast quasi-2D schematisation not only allows enough simulations to be performed to generate localised and system-wide risk estimates, it also provides the opportunity to assess negative system behaviour effects due to defence failures in the floodplain domain.

## 2. Load Interdependencies in The Netherlands

Since the 16th century, the compartmentalisation of floodplain areas in the Netherlands has developed primary defences that protect polders from the rivers and sea, and secondary defences that divide the polders into smaller regions [31]. The primary defences consist of multiple dike rings that have been assigned higher and higher protection standards over the years, due to the occurrence of extreme events and increased economic and societal exposure [32]. New protection standards to which the primary defences must conform have recently been imposed under Dutch law [33]; however, it is known that most of the defences do not currently adhere to these standards.

The most recent estimate of current protection levels are given by the VNK2 project [34], however, the calculated failure probabilities are generally considered to be conservative. Within the study, the probability of breaches along defined sections of dike was used to estimate the overall failure probability of 'trajects' or segments of the dike ring defences. The VNK2 project only accounted for load interdependency effects between dike rings 14, 15 and 44 (see Figure 1), in effect treating this highly developed area as a single dike ring [35]. However, investigations into load interdependencies by Delft Hydraulics [36] suggest that negative effects are likely to be significant in a number of regions not addressed in the study, three of which are highlighted in the present study.

The first location for which spatial aspects of load interdependencies may be relevant is in dike ring 43 (the 'Betuwe', see Figure 1, location A). The region is shown to be vulnerable to floods both economically [37] and with respect to loss of life [38]. Simulations of breach flows into this dike ring demonstrate how secondary defences delay and compartmentalise the flood waters, causing high water depths upstream of these defences. Studies suggest this effect reduces overall economic risk and further 'compartmentalisation' of the region would likely further reduce risk [39]. Flood water can overflow back into the river system downstream in the dike ring, however failure of this system would likely result in a cascading or domino effect of flows into dike ring 16 ('Alblasserwaard') [40].

Another potential floodplain shortcut highlighted by the authors of [36] is dike ring 41, or 'Land van Maas en Waal' (Figure 1, location B). As the name suggests, this region sits between the Waal and Meuse rivers, which converge to within 1 km of each other at the Western end of the dike ring. The dikes on the Meuse are generally lower than those of the Waal in the regions of dike rings 41 and 40 (see Figure 1 below). This, together with the Meuse's smaller capacity means that large breach flows originating from the Waal could increase flood risk downstream on the Meuse. A probabilistic computational framework for system behaviour analysis of this area was described by Courage et al. [41]. The authors concluded that load interdependencies are highly significant in the area, and that a framework encompassing the entire system was required for further analysis.

The potential for cascading and shortcutting of breach flows originating on the right bank of the German Rhine and propagating down the IJssel (through dike rings 48–53, see Figure 1, location C) has been addressed in studies by Klerk [1] and Bomers et al. [42]. In both studies, a large redistribution of risk was observed. During high flows on the Rhine, bifurcation control structures convey only ~1/9 of the Rhine flow towards the IJssel, due to the limited capacity of this branch. However, breaches on the right bank of the Rhine, upstream of the bifurcations, have the potential to increase this flow beyond the capacity of the river, should the flows rejoin the system.

This paper analyses load interdependencies in the Dutch river system that include the potential for spatial inundation effects such as cascading and shortcutting. Scenarios are evaluated to model the effects of polder-side and regional defence failures, and the results from the three locations described above (dike ring 41, dike ring 43 and the IJssel valley) are analysed in detail. While load interdependency analysis that include the potential for negative interdependencies have been applied to isolated areas in the Netherlands, it was not previously possible to quantify the effect on overall hazard in the system. Furthermore, by developing scenarios that allow for polder-side and regional defence failures, the effects of a nonstatic floodplain domain can be assessed.

## 3. Methodology

### 3.1. Case Study

The case study is based on the lower Rhine and Meuse Rivers in the Netherlands. The floodplains of this region have been compartmentalised into a system of dike rings in which roughly 67% of the population of the Netherlands (17 million people) live. The area has been schematised in a calibrated Sobek 3 model [30] (see Figure 1). Downstream, stage-discharge boundary conditions are used on the Meuse, and on the three Rhine branches of the Waal, Lek and IJssel. These locations are considered to be upstream of where tidal influences are dominant in the system. The upstream boundary conditions use available hydraulic load distribution data from the 'GRADE' (Generator of Rainfall and discharge Extremes) project [43] for both the Meuse and Rhine rivers. Smaller tributaries are ignored or modelled as steady state contributions.

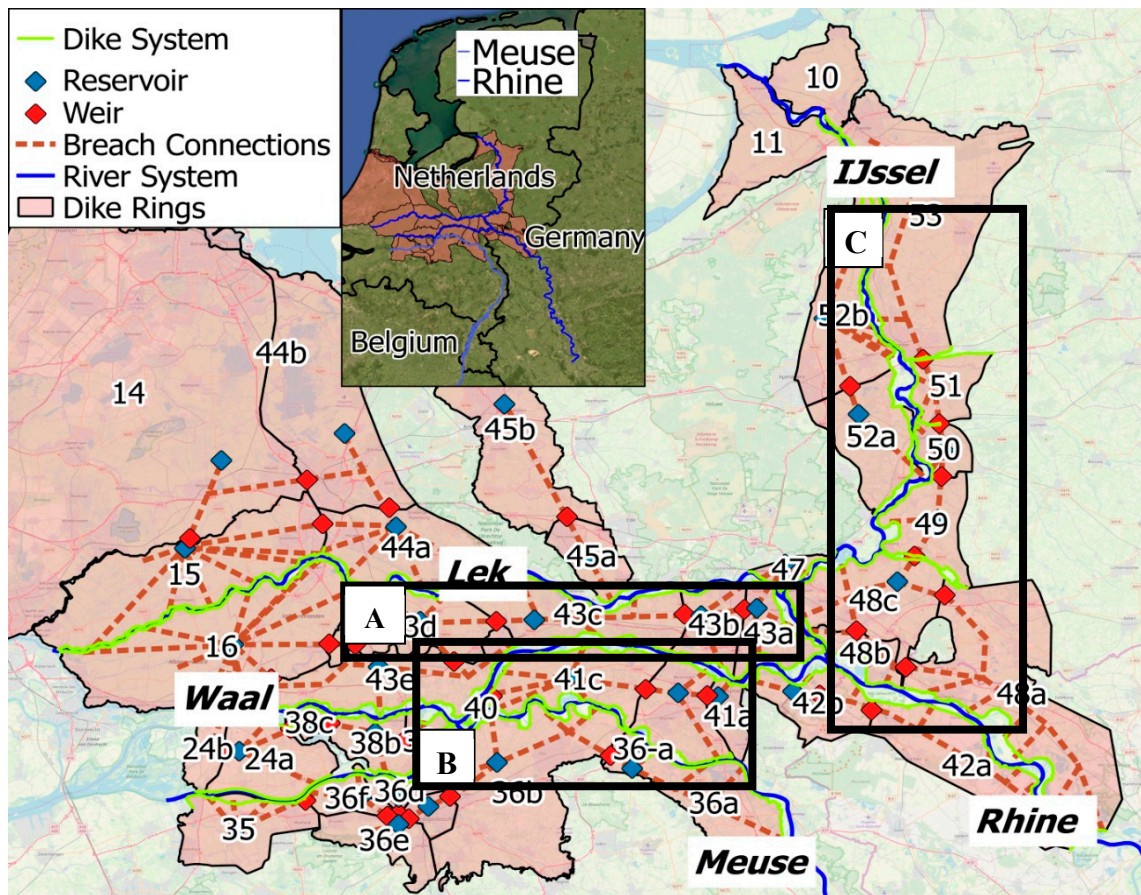

**Figure 1.** Schematised river and dike system for the Netherlands, as well as quasi-2d floodplain schematisation. The numbers refer to the dike ring IDs used in the model. Inset: Relative location of case study. Three specific locations discussed in the results are highlighted in black (A–C).

As the risk of pluvial flooding is small relative to fluvial, floodplain inundation is only possible from the rivers. Potential breach locations and the dike ring sections they represent are defined as per the VNK2 study, with the addition of 12 locations in Germany, where dike rings 42 and 48 extend across the Dutch–German border. These latter breach locations are defined as per inter-agency reports on the dike rings [44]. Other potential breaches further upstream on the Rhine are likely to be smaller and contained within the floodplain valley; however, the effect of these breaches on the incoming floodwave is a limitation of the study.

Historical breaching events such as floods in 1805 and 1926 have been suggested as precedents for cascading and shortcutting flows [45,46]; however, too much has changed in the river and floodplains to consider these events for validation of 2D schema. Instead, the present schematisation was validated using coupled 1D-2D breach simulations from available models and a repository of existing 1D/2D simulations [47].

In the Sobek model, once a breach occurs, it enters the floodplain domain, which is delineated by the dike rings (Figure 1). To reduce the computation time required, the flow is schematised as either 1D or 0D in the floodplains, and the available 1D/2D breach simulations are used to delineate, schematise and calibrate the floodplain domain.

An example of this process for dike ring 43 can be seen in Figure 2, showing the inundation calculated over time by a particular breach simulation, as well as the final schematisation used in the present study. Here, the breach flow is considered to behave like a series of connected reservoirs (i.e., 0D) rather than a floodplain flow route (1D). Areas where the floodplain flow builds up before overflowing into a new region are in this case delineated by major roads and secondary defences. Breaches from the river fill a shared reservoir representing the retention capacity of the region in relation to the water depth, which is calculated through a GIS analysis of that area. The location of overflows that connects the compartments is schematised with a weir, which can be adjusted dynamically. The locations of these reservoirs and weirs are shown in Figure 1. The 0D schematisation was applied almost unilaterally to the compartments delineated in this analysis, apart from upstream regions in which the steeper topography was better schematised by 1D flow route.

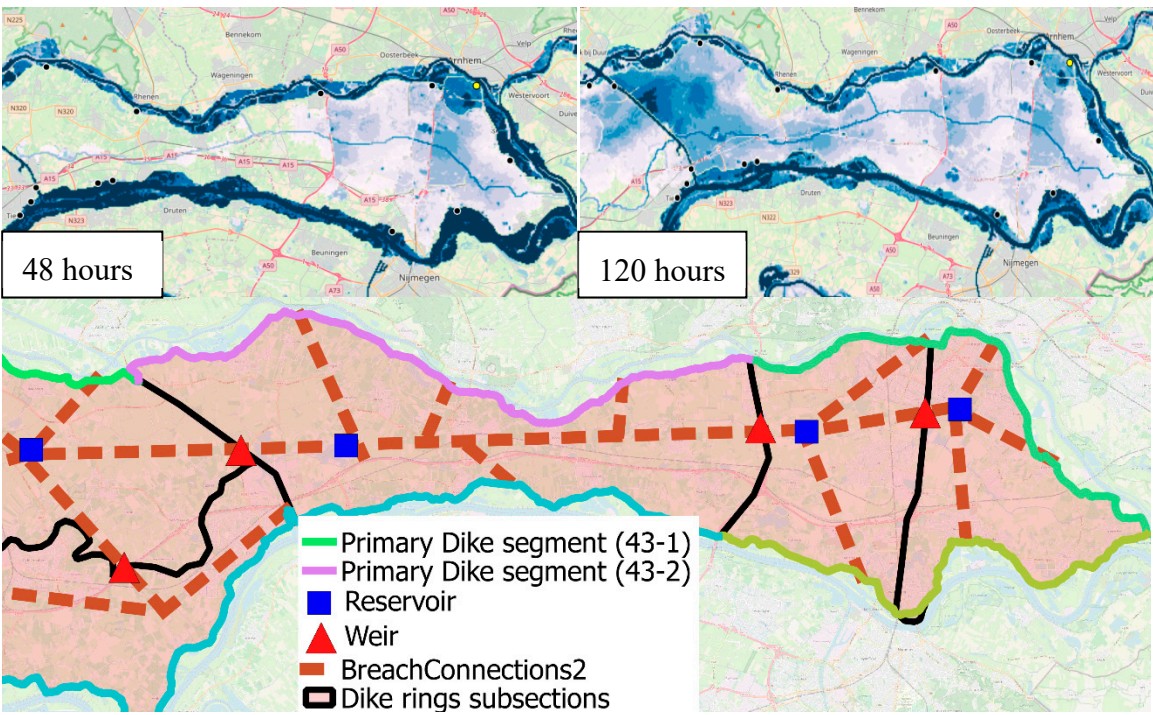

**Figure 2.** Top panels: Flooding from the Lek to dike ring 43 over time. Bottom panel: resulting delineation and schematisation of dike ring 43 based on flood simulations.

The 2D simulations were also used to benchmark the quasi-2D model. At a given location, breach outflow discharge hydrographs from the 2D model are reproduced in the quasi-2D model by imposing the same floodwave and breach growth conditions. Then the water-level time series from particular 2D cells at downstream locations in the compartments are compared to the water-level time series from the reservoirs used in the quasi-2D model. These comparisons showed the level and timing of inundation to be sufficiently approximated by the quasi-2D model to represent flows and interactions

in the system. The use of the quasi-2D model reduces computation time from about 4 h to 8 min on an Intel Core i5 laptop with 8 Gb of RAM.

The maximum inundation volume stored in the compartments in any simulation is here used as the metric to indicate hazard. Even though the quasi-2D model was validated with respect to timing and inundation depths, maximum volume was used due as it is more consistently accurate and demonstrates interactions relatively clearly. In general, the use of a quasi-2D model prevents other important hazard variables (such as velocity [48,49]) from being obtained. This is a significant limitation of the study that could only be overcome with the use of a more complex 2D model.

### 3.2. Computational Framework and Scenarios

The computational framework used to assess the spatial risk is adapted from [15]. A Monte Carlo analysis is performed in which multiple event parameters are sampled from distributions of load (discharge peak and wave shape) and dike strengths (fragility functions). The hydraulic load distributions are obtained from the GRADE database [43], which produces 50,000 yrs of flow data using a weather generator. Both distributions are tabular or relative frequency distributions; however, the wave shape parameter relates to a complete hydrograph that is then scaled according to the sampled peak. This allows for the variability in the duration of flood waves to be included in the simulations. Correlation between sampled events on the river was introduced using a correlation factor of 0.9, which is taken from a report on the dependence of the Meuse and Rhine Rivers by Diermanse [50]. The dike strengths are sampled from fragility functions that are discussed below.

The events are then simulated in the hydrodynamic model to assess the resulting water levels, discharges, breaches and inundation volumes. Importance sampling is used to generate 16,000 simulations for each scenario (see Table 1), which was sufficient to ensure convergence of the failure probabilities at each dike section. The outcomes of all simulated events are combined to derive hazard characteristics such as failure probabilities, discharge and water level distributions at various locations and floodplain inundation statistics.

Four implemented scenarios of this framework are assessed to evaluate the effects of inter-riverine flow (caused by polder-side breaching) and inter-regional flows (caused by breaches between dike rings). In all scenarios, flow can occur over defences (Table 1).

**Table 1.** Scenarios used in analysis, indicating which failure mechanisms are used in each scenario.

| Scenario Name | River Breach | Regional Breach | Polder-Side Breach |
|:---:|:---:|:---:|:---:|
| NoSys | ✗ | ✗ | ✗ |
| RivBr | ✓ | ✗ | ✗ |
| RegBr | ✓ | ✓ | ✗ |
| PolBr | ✓ | ✓ | ✓ |

In the 'NoSys' scenario, no system behaviour occurs as water does not leave the 1D domain; however, the breaches that would have occurred are recorded for the purposes of comparison. In the 'RivBr' scenario, breaches can occur from the river but do not occur between dike rings and regions, or from the polder back into the river. In the 'RegBr' scenario breaches can occur between regions and dike rings, and in the final scenario ('PolBr'), breaches can also occur from polders back into the river. In all scenarios overflow can occur between domains, even if a defence has not been breached.

For river breaches, predetermined breach locations are used to represent a section of dike with a given failure probability. These failure probabilities are based on the VNK2 assessment for each dike 'traject' or segment. At each breach location, previously developed fragility surfaces [15], are used to represent the failure probabilities as well as their associated uncertainty in terms of water level and duration of exceedance of that water level. These surfaces are generated combining the three main failure mechanisms of overtopping, piping and macrostability. The water level uncertainty is based on the amalgamation of fragility curves [51], while the duration component is elicited from

expert opinion [15]. Breach triggering thresholds for each Monte Carlo simulation are sampled from the fragility surfaces, which allow for stochastic breaches before and after the peak of the floodwave. The growth of the breaches is defined using the formula of Verheij and van der Knaap [22], using the default parameters for initial breach width (10 m) and critical flow velocity (0.2 m/s).

In the available 2D breach simulations, the topography of the floodplain is considered static (apart from the initial dike breach). In low-lying and compartmentalised regions, however, barriers to flow may succumb to the water pressure and overtopping volumes from the inundated floodplain. Therefore, the possibility of their breaching is included in 'RegBr'. Due to a lack of research into this type of breaching, this is assumed to be at the moment of initial overtopping, when the formula of Verheij and van der Knaap [22] is again used to estimate breach growth.

In the 2D simulations, large pressures and overtopping volumes are also observed from the polder side of the dike rings to the rivers, which may cause breaching in this direction. The 'PolBr' scenario, therefore, also accounts for this failure type. In this scenario, polder-side breaching can in theory occur at any location where a river breach is possible, but only practically occurs at locations where flood volumes accumulate and exert pressure on the polder-side slope (i.e., at the downstream end of a dike ring). The method to account for these failure types has been adapted from Klerk [1] and heuristic conversations with dike stability experts. For the fragility of river side breaching, breaches can occur due to overtopping and macrostability, whereas piping is ignored due to the likely presence of high water pressures on both sides of the dike. Overtopping failures occur when the floodplain levels surpass dike height for 3 h, whereas macrostability failures occur when the floodplain levels are 3 m higher than the river levels for a period of 12 h or more. The results from all scenarios are shown and discussed in the following section, for the entire system and for three specific locations of interest.

## 4. Results and Discussion

### 4.1. Overall System

The clearest impact of load interdependencies is observed when assessing extreme events that have the potential to cause more than one breach in the system. On the left side of Figure 3, the expected number of breaches in an event of a given return period are shown for each scenario (note that in the 'NoSys' scenario, breaches that would have breached are recorded, but no flow leaves the river system). The number of breaches is seen to be higher in the 'NoSys' scenario for return periods >500 years. The right side of Figure 3 shows the net flow of volume out of all breach locations. This net volume is slightly reduced when flow can breach from the polder-side in the 'PolBr' scenario. The 'NoSys 'scenario does have some inundation due to overtopping at extreme events, but this is several orders of magnitude smaller than the other scenarios.

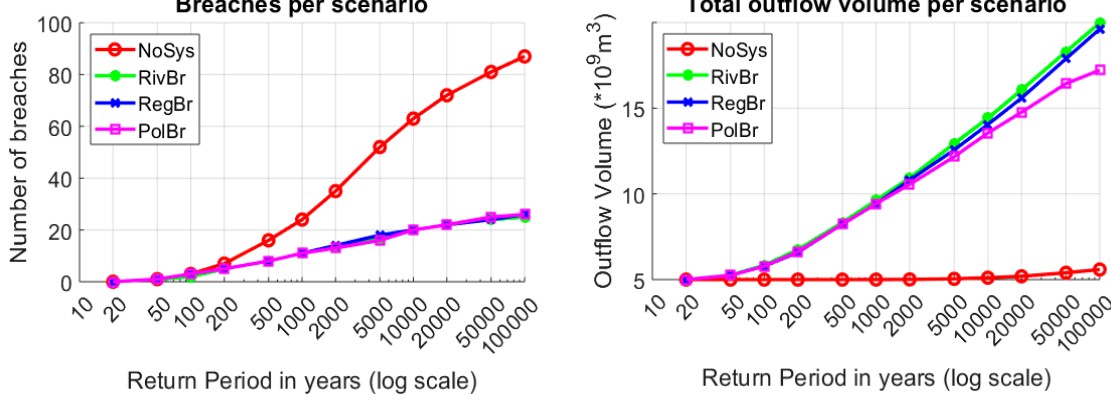

**Figure 3.** Exceedance probabilities in the entire system for number of breaches (**left**) and total breach outflow, (**right**).

These results are similar to those observed in the works by the authors of [21] and [15], in which the lower downstream discharges and failure probabilities due to load interdependencies are also highlighted. However, the 1D models used in those studies make it difficult to ascertain local system behaviour effects caused by floodplain flow. Three localised system behaviour interactions are therefore examined in detail below. The locations are as discussed earlier, i.e., dike ring 43, dike ring 41 and dike rings 48–53 (The IJssel valley), (Figure 1).

## 4.2. A: Dike Ring 43

Dike ring 43 is enclosed on the west by a regional defence to dike ring 16 (Figure 4). In the quasi-2D model, overflow is possible from subsections 43d and 43e into dike ring 16, and in the 'PolBr' and 'RegBr' scenarios this regional defence can be breached. The expected inundation volumes for various return periods in these regions are shown in Figure 4. From these graphs it can be seen that regional failures increase the expected inundation in dike ring 16, while lowering inundation volumes in 43d and 43e. While the volume changes are relatively small, these regional failures cause inundation in dike ring 16 for events with a shorter return period, which is likely to increase the overall risk.

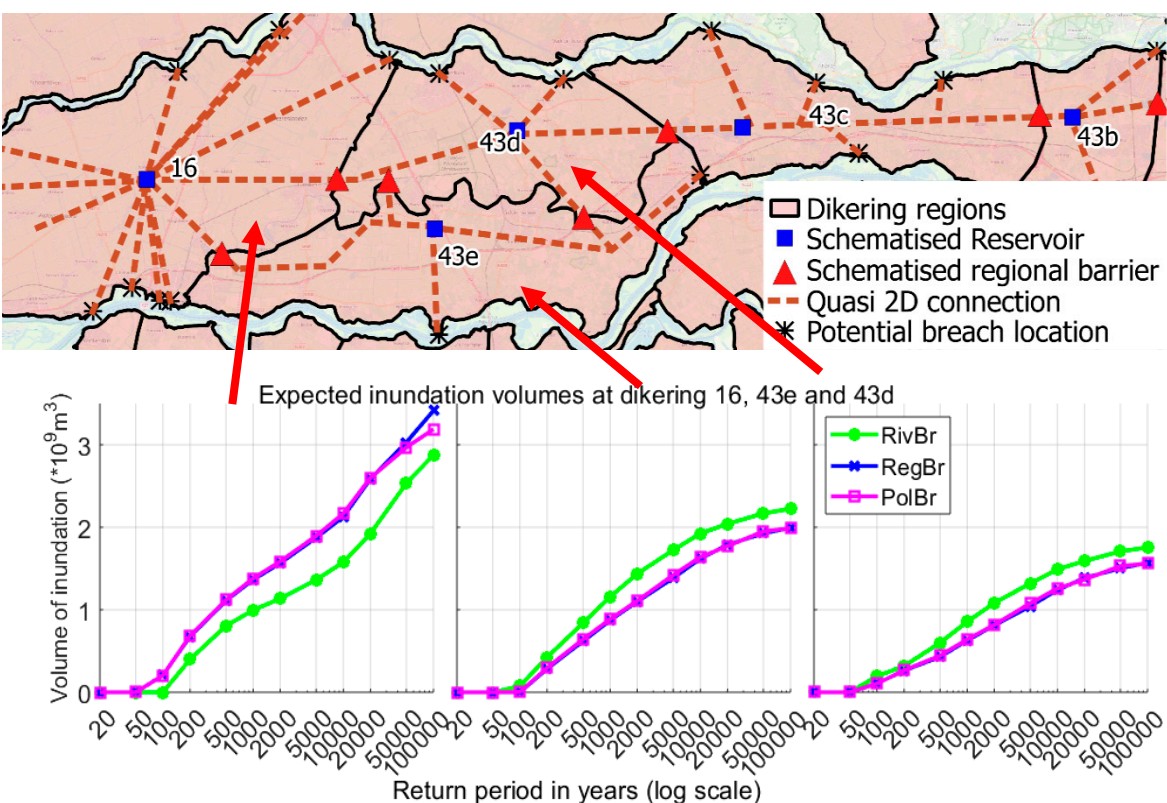

**Figure 4.** Breaching of regional defences in dike ring 43 (Area A). Top: Close up of dike ring 43. Bottom: expected inundation volumes at compartments 16, 43e and 43d.

## 4.3. B: Dike Ring 41

'Shortcutting' (i.e., breach flows that rejoin a river system through the polder) has the potential to occur at dike ring 41, which lies between the Waal and Meuse rivers (Figure 5). Of the simulations in which the dike ring is inundated, breaching of dike segment 41-2 on the Waal is most often the cause. The resulting inundation volumes are usually high, as the Waal takes a large portion (~2/3) of the discharge from the Rhine. The expected inundation volumes for compartment 41c are given in Figure 5, and the scenario with the potential for polder-side breaching (PolBr) is seen to have lower expected volumes during extreme events than scenarios without polder-side breaching. A consequence of the polder-side breaching into the Meuse is that the expected discharges downstream are affected. For the

PolBr scenario, extreme discharges downstream in the Meuse are comparable to the scenario in which no breaching occurs anywhere (NoSys).

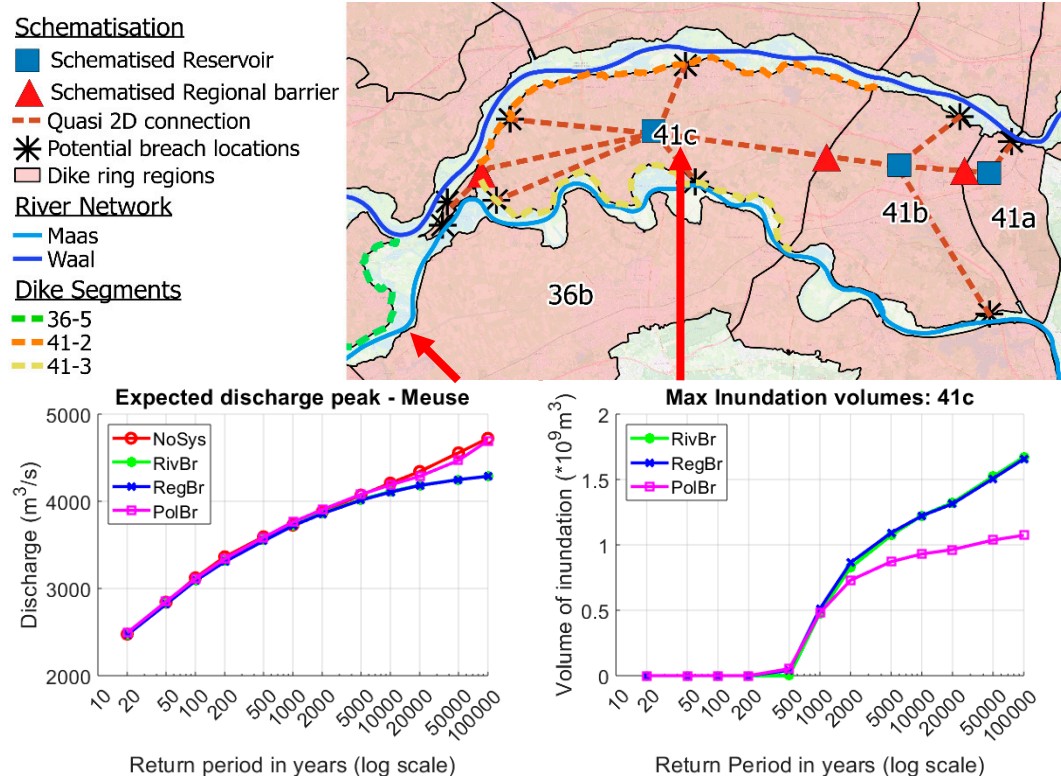

**Figure 5.** Example of shortcutting from Waal to Meuse. (**Top**) Expanded version of location B from Figure 1. Inundation volumes are seen to reduce in compartment 41c for scenario 'PolBr' (**Bottom right**). This increases the expected discharge downstream in the Meuse (**Bottom left**).

### 4.4. IJssel Valley

Analysis of 2D breaching simulations show that flood flows from breaches that occur on the right bank of the German Rhine generally flow toward the IJssel. In Figure 6, compartment 48c is seen to have higher expected inundation volumes for extreme events in the scenario in which failure of regional defences is not possible (RivBr). In the other scenarios, failure of the regional defence from 48c to 49 occurs, reducing the expected volume. In the simulations, flows cascade down the IJssel valley through dike rings 50 and 51. However, large pressures build up in these compartments, and in the scenario in which polder-side failures can occur (PolBr), the expected volumes are reduced in compartments downstream (such as dike ring 51), due to polder-side breaching.

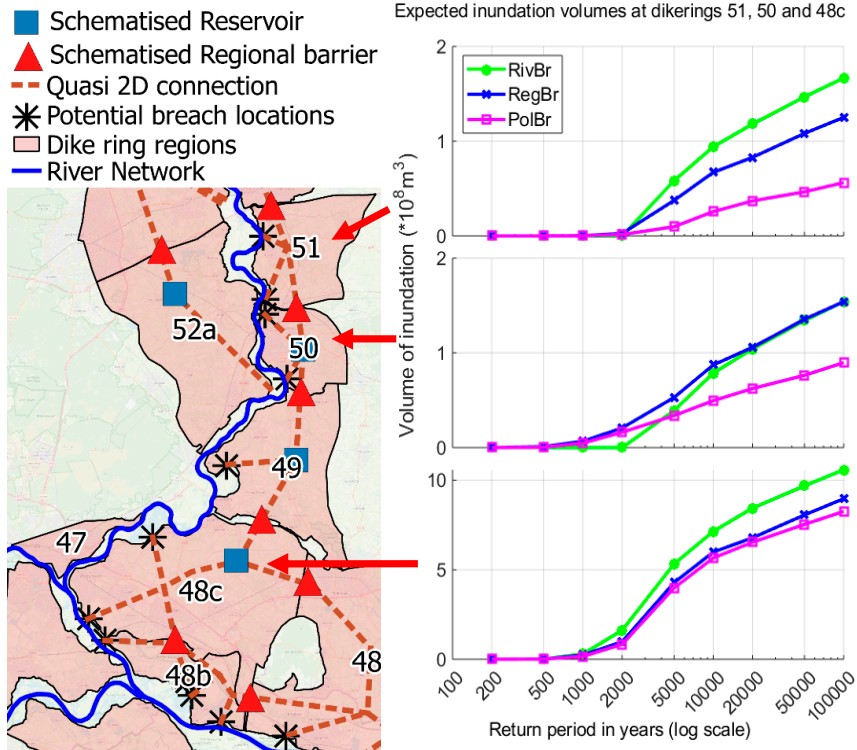

**Figure 6.** (**Left**) Expanded version of location C from Figure 1. (**Right**) Expected inundation volumes in dike ring compartments 48c, 50 and 51 for all scenarios. N.B. *Y*-axis scale changes in bottom graph.

## 5. Conclusions

For many protected river systems, the overall and local flood risks computed from large-scale analyses have been shown to be dependent on the 'load interdependency' effects that are observed when dike breaching is accounted for. The probabilistic computation required for such studies can preclude the use of sophisticated 2D models, often resulting in certain river–dike–floodplain interactions to be overlooked in the analysis. Even with 2D models, static topographies do not allow for floodplain interactions such as regional and polder-side breaching.

As the floodplains of many large protected lowland river systems are compartmentalised, fast quasi-2d hydrodynamic models are often a reasonable alternative to estimate breach flow inundations. Furthermore, these models can include breaching of secondary defences that delineate the compartments. This paper conducted a load interdependency analysis of the Dutch river system using a fast quasi-2D model that allowed for system-wide and localised risk estimates to be generated and 'negative load interdependencies' to be demonstrated. The 2D domain was calibrated on existing breach simulations, and four scenarios were investigated to evaluate the effects of river-side, regional and polder-side breaching.

The case-study results showed that load interdependencies can have a significant impact on hydraulic load distributions and thus failure probabilities in the system, which is in agreement with results from the works of the authors of [1,15,21]. The addition of a fast quasi-2D model to the 1D domain allowed for useful statistics (such as expected inundation volumes), to be estimated at each region, and highlighted areas in which inundation would cause significant pressures on secondary/regional defences or on the polder-side of primary defences. Failure of these defences was explored in two scenarios, and in many regions these failures have a large impact on the volume, level and timing of flood hazards. Specifically, the model highlighted the possibility of: 'shortcutting' and 'cascading' of flows in the regions of; the German Rhine to the IJssel, dike ring 41 and dike ring 43.

The study is limited in a number of areas. While fast, the quasi-2D approach does not allow for certain hazard variables to be calculated that may contribute to risk, such as flow velocities.

The coincidence of river discharges and storm surges and tidal influences is also not currently included. Various assumptions were required for the analysis, such as the simplified thresholds for failure of regional and polder-side defences. The failure probability of the dike sections assumed in the paper (from the VNK2 study, [34]) are considered to be too high, and many of these sections are currently being reassessed in light of the new safety standards.

These new standards are optimised based on (among other variables) expected economic damage from breaching, but do not include many of the interactions explored in this study. Therefore, calculating the expected economic damage and loss of life resulting from a system behaviour analysis under the new safety standards could be a worthwhile future research for the case study. In general, the approach developed can be used to evaluate risk and develop scenarios for many protected floodplain systems such as the Elbe, Po, Mekong and Mississippi rivers. The results could be of use for decision-making in many large-scale flood risk management applications, for example, when evaluating compartmentalisation or detention area mitigation strategies.

**Author Contributions:** Conceptualization, A.C., W.J.K. and K.M.D.B.; methodology, A.C.; validation, A.C.; writing—original draft preparation, A.C.; writing—review and editing, K.M.D.B., W.J.K. and M.K.; visualization, A.C.; supervision, K.M.D.B. and M.K.

**Funding:** This project has received funding from the European Union's Horizon 2020 research and innovation programme under the Marie Skłodowska-Curie grant agreement No 676027.

**Conflicts of Interest:** The authors declare no conflict of interest.

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
