# Peer review of "Large Scale Flood Hazard Analysis by Including Defence Failures on the Dutch River System"

_water, doi:10.3390/w11081732_

Round 1
Reviewer 1 Report
This is a well written paper and valuable area of research. I have provided some comments and edits in the PDF. Here are some additional comments:
The writing needs some help with punctuation. Please add the research questions and novelties (unique aspects) in the last paragraph of Introduction right after the objective sentence. This would make it easy for the reader to understand the study contribution to the body of literature. Please present the number of people living in the case study. How many Monte Carlo trials were settled? Did you ensure that this number is sufficient for numerical convergence? Please provide details on the probability distributions (of load—discharge peak and waveshape—and dike strengths—fragility functions) in the Monte Carlo analysis and explain why you selected these distributions. Was any correlation considered between the stochastic parameters in the Monte Carlo simulation? Did you consider sunny-day (no rainfall) conditions for all the four scenarios? It would be useful to incorporate some rainfall-drive flooding to your analyses. While using flood depth in hazard estimation is common, you should discuss why other characteristics (e.g., velocity and duration) were not investigated? You should also discuss how well the model could predict these characteristics (compared to the 2D benchmark model). This, at least, needs to be discussed as a limitation. See and acknowledge past studies, including Kreibich et al. (2009), Dang et al. (2011), Qi & Altinakar (2011) and Ahmadisharaf et al. (2015). Were return periods of breach outflow volume and discharge determined based on Monte Carlo simulations? Given that these are based on model simulations and not observations, is this appropriate to be called return period like what is done for observed peak flow or storm? It would be helpful to add outflow peak plot into Figure 3. Please present the simulation run times and computer specifications by both quasi-2D and the benchmark 2D models. Why there is no comparison on spatially distributed flood characteristics (e.g., max depth/velocity, duration and arrival time) and time-dependent characteristics (e.g., mean depth/velocity and rate of rise)? The latter could be done as you applied an unsteady model. Please explicitly discuss how your study helps risk mitigation in FRM decision making phase. Please expand the study limitations and the recommendations for future research.

Author Response
Dear Reviewer,
Thank you very much for your comments and advice. The authors have written a new draft (to be uploaded shortly), based on these comments, as well as those from the other reviewer.
Please see attachment for this response.
Regards,
Authors

Reviewer 2 Report
In the manuscript the authors present the results of a case study of load interdependency analysis in the Netherlands combined with fast 2D hydraulic modelling to produce hazard estimates under various defence failure scenarios. These scenarios are evaluated to model the effects of polder-side and regional defences, and the results are analysed for three selected locations in the region.
Although the quality of the paper is overall good, I cannot find a substantial innovation in it, as the authors merely applied established and/or existing methods/models to a case study. This could not be necessarily a problem; however, if we want to consider the present manuscript as a “case-study paper”, I think it would require some additional effort, especially regarding a more detail in the description of the methodological approach and the consequent discussion of the results. Indeed, the authors state in P5.L187-188 that “A Monte Carlo analysis is performed in which multiple event parameters are sampled from distributions of load (discharge peak and waveshape) and dike strengths (fragility functions)”, without providing any other additional information and further discussion.
Minor comments:
P1.L38: “:” instead of “;”.
P2.L16: missing “i” in “interdependencies”.
P2.L62: “flow is solved”? Please rephrase.
P2.L71: insert comma after “capacity”.
P3.L102: please include reference to Figure 1, when talking about dike ring 14, 15 and 44.
P3.L117: please rephrase.
P3. L126: delete “in” after “studies”.
P3.L128: “direct”? Maybe “convey”?
P3.L129: “before” may be changed with “upstream”.
P3.L129-130: consider rephrasing of this sentence
P4.Figure 1: explain in the figure caption that the numbers shown in the figure refer to dike rings ID.
P6.L220: missing “is” after “[22]” and missing “to” before “estimate”.
P7.L254: missing reference.
P10.L312: “:” instead of “;”.
Author Response

(The authors gave the same response as above.)

Round 2
Reviewer 1 Report
Most of my comments were addressed. I am repeating the PDF comments that were not visible by the authors here and some additional comments:
‘flood risk management’ instead of ‘Flood Risk Management’. There should be a space between numbers and units (e.g., 1.2 km and 50,000 yrs) throughout the text. Please spell out all the abbreviations the very first time they appear in the text (e.g., one-dimensional [1D], two-dimensional [2D] and etc.). L87: Please specify section/subsection instead of saying ‘below’. L89: Please briefly discuss what the scenarios are here. Please briefly discuss what a 0D model is (where it appears first in the text). Please discuss why a correlation coefficient of 0.9 was used. What does the citation refer to? Is it for the same application and study area? Add your response to my comment ‘Did you consider sunny-day (no rainfall) conditions for all the four scenarios? It would be useful to incorporate some rainfall-drive flooding to your analyses.’ somewhere in the manuscript. You should still discuss why there is no analysis/comparison on spatially distributed flood characteristics (e.g., max depth/velocity, duration and arrival time)? Please explicitly discuss how your study helps risk mitigation in FRM decision making phase. Despite your response, I do not see some discussion added in the revised manuscript. Please address the edits and comments in the PDF.

Author Response
Thank you for your re-review. See attached response sheet.

Reviewer 2 Report
Authors have mostly addressed the comments raised by me and by the other reviewer.
Please note only a very minor correction in P5.L200: change "for" with "that".
Author Response

(The authors gave the same response as above.)
